# Deciphering the Roles of Interspace and Controlled Disorder in the Bactericidal Properties of Nanopatterns against *Staphylococcus aureus*

**DOI:** 10.3390/nano10020347

**Published:** 2020-02-18

**Authors:** Khashayar Modaresifar, Lorenzo B. Kunkels, Mahya Ganjian, Nazli Tümer, Cornelis W. Hagen, Linda G. Otten, Peter-Leon Hagedoorn, Livia Angeloni, Murali K. Ghatkesar, Lidy E. Fratila-Apachitei, Amir A. Zadpoor

**Affiliations:** 1Department of Biomechanical Engineering, Faculty of Mechanical, Maritime, and Materials Engineering, Delft University of Technology, 2628CD Delft, The Netherlandsl.angeloni@tudelft.nl (L.A.);; 2Department of Imaging Physics, Faculty of Applied Sciences, Delft University of Technology, 2628CJ Delft, The Netherlands; 3Department of Biotechnology, Faculty of Applied Sciences, Delft University of Technology, 2626HZ Delft, The Netherlands; 4Department of Precision and Microsystems Engineering, Faculty of Mechanical, Maritime, and Materials Engineering, Delft University of Technology, 2628CD Delft, The Netherlands; m.k.ghatkesar@tudelft.nl

**Keywords:** nanoscale additive manufacturing, surface nanopatterns, antibacterial effects, controlled disorder, interspace

## Abstract

Recent progress in nano-/micro-fabrication techniques has paved the way for the emergence of synthetic bactericidal patterned surfaces that are capable of killing the bacteria via mechanical mechanisms. Different design parameters are known to affect the bactericidal activity of nanopatterns. Evaluating the effects of each parameter, isolated from the others, requires systematic studies. Here, we systematically assessed the effects of the interspacing and disordered arrangement of nanopillars on the bactericidal properties of nanopatterned surfaces. Electron beam induced deposition (EBID) was used to additively manufacture nanopatterns with precisely controlled dimensions (i.e., a height of 190 nm, a diameter of 80 nm, and interspaces of 100, 170, 300, and 500 nm) as well as disordered versions of them. The killing efficiency of the nanopatterns against Gram-positive *Staphylococcus aureus* bacteria increased by decreasing the interspace, achieving the highest efficiency of 62 ± 23% on the nanopatterns with 100 nm interspacing. By comparison, the disordered nanopatterns did not influence the killing efficiency significantly, as compared to their ordered correspondents. Direct penetration of nanopatterns into the bacterial cell wall was identified as the killing mechanism according to cross-sectional views, which is consistent with previous studies. The findings indicate that future studies aimed at optimizing the design of nanopatterns should focus on the interspacing as an important parameter affecting the bactericidal properties. In combination with controlled disorder, nanopatterns with contrary effects on bacterial and mammalian cells may be developed.

## 1. Introduction

An increasing number of orthopedic implants are being implanted every year [1], resulting in a growing number of implant-associated infections (IAIs). Despite all efforts involved in preventing infections in clinical settings, IAIs still occur and are recognized as one of the most prevalent causes of the failure of orthopedic implants [2,3]. Such infections usually necessitate either revision surgeries or the prolonged administration of antibiotics, which diminishes the patients’ quality of life, causes major side effects, significantly increases the healthcare costs, and could lead to patient morbidity or even mortality [3,4]. *Staphylococci* bacteria are the most widespread infectious pathogens involved in IAIs [5]. While *Staphylococcus aureus* accounts for 20–30% of IAIs following fracture fixation and prosthetic joint infections [5,6,7], antibiotic treatment could act as a double-edged sword, particularly given the growing crisis of antibacterial resistance leading to the evolution of antibiotic-resistant species like Methicillin-resistant *staphylococcus aureus* (MRSA) [8]. Moreover, recent reports have shown that in addition to antibiotics, bacteria can develop resistance against other types of killing agents, such as silver nanoparticles [9]. Therefore, alternative approaches to the prevention of IAIs including those based on physical forces should be more seriously considered. In such approaches, specifically designed surface features kill the bacteria that reach the implant surface [10].

The recent progress in nano-/micro-fabrication techniques have made it feasible to develop surfaces ornamented with geometrical features (e.g., pillars) whose arbitrary shapes, sizes, and arrangements are precisely controlled [11]. Eukaryotic and prokaryotic cells are known to interact with these patterns in different ways ending up with distinct, and even contrary, cellular responses [12,13]. There are pieces of evidence suggesting that mechanobiological pathways trigger and control these responses [12,14,15]. At the same time, nature offers great examples of surfaces with nanoscale features (nanopatterns) that leave bacterial cells with no choice of response but death [10,16,17]. Therefore, studying the interactions between different types of cells and surface nanopatterns is of high interest, because, unlike larger length scales, nanopatterns can affect individual cell receptors, which stand first in the line for the transduction of mechanical signals [14,18].

Inspired by nature, many synthetic replicas of naturally occurring bactericidal nanopatterns have been designed, fabricated, and tested against a wide variety of bacterial strains [17,19]. Several design parameters such as the shape, dimensions (height, diameter, and the interspace between them), and arrangement highly influence the response of bacterial cells to surface nanopatterns [12,20]. For example, a specific range of dimensions is known to induce bactericidal properties (i.e., 100 nm < height < 900 nm; 20 nm < diameter < 207 nm; 9 nm < interspacing < 380 nm) [20].

The limited number of systematic studies has made it difficult to draw concrete conclusions regarding the isolated effects of each design parameter (i.e., height, diameter, or interspacing) on the bactericidal properties of surfaces. Similarly, while extensive data is available regarding the effects of disordered nanopatterns on the response of mammalian cells [21,22,23], a limited number of reports can be found on how disordered arrangement can affect the bactericidal properties of nanopatterns [24].

In the present study, we aimed to study the isolated effects of one specific design parameter (i.e., interspacing) as well as controlled disorder on the functionality of a bactericidal nanopattern. Pillar-shaped nanopatterns with an approximate height of 190 nm, a diameter of 80 nm, and an interspacing of 170 nm were chosen as the reference bactericidal nanopattern, which has been shown to be effective against both Gram-positive and Gram-negative bacteria [25]. Keeping the height and diameter constant, the interspacing of the nanopillars was changed to 100, 300, and 500 nm to create new nanopatterns. These values cover the full range of possible interspacing including one that was larger than all previously reported ones (i.e., 500 nm), one close to the maximum value reported before (i.e., 300 nm), and one smaller than the majority of the previous studies (i.e., 100 nm) [20]. The controlled disorder was the other studied parameter, creating a variant to each of the four abovementioned nanopatterns. Furthermore, *S. aureus* was used as the study organism because of its prevalence in IAIs. We used electron beam induced deposition (EBID) as a nanoscale additive manufacturing (3D printing) technique to fabricate the nanopatterns due to its high precision and controllability that make it an unrivaled single-step method for the direct printing of 3D surface physical features at the sub-10 nm nanoscale [25,26,27].

## 2. Materials and Methods

### 2.1. Nanopatterns Design, Fabrication, and Characterization

To introduce controlled disorder to the nanopatterns, the maximum distance at which nanopillars did not intersect when displaced towards each other was set as the maximum disorder distance. Having set the diameter of nanopatterns to 80 nm, the maximum disorder distance was defined for each nanopattern to be half of the difference between the interspacing and the diameter. Since such a small level of disorder does not substantially change the arrangement of the nanopatterns with an interspacing of 100 nm, the effects of disorder were not studied for that particular level of interspacing. The following nanopatterns were, therefore, included in the study: ordered nanopatterns with interspacings of 100, 170, 300, and 500 nm as well as disordered nanopatterns with interspacings of 170, 300, and 500 nm. We will call those patterns 100 *O*, 170 *O*, 170 *D*, 300 *O*, 300 *D*, 500 *O*, and 500 *D* where the first number corresponds to the interspacing of the nanopillars followed by a latter indicating whether the arrangement of nanopillars has been fully ordered (*O*) or included controlled disorder (*D*).

EBID was used to fabricate the desired nanopatterns on silicon substrates as described before [25]. Briefly, 1 × 1 cm^2^ samples were prepared by cutting double-sided polished 4-inch (diameter = 10.16 cm) silicon wafers (thickness = 525 ± 25 µm, p-type), cleaning with nitric acid, and rinsing with deionized water subsequently.

A Helios Nano Lab 650 scanning electron microscope (SEM) (FEI company, Hillsboro, OR, USA) equipped with the apparatus required for EBID was used to create three nanopatterned areas of 20 × 20 µm^2^ per specimen (Figure 1d). The precursor gas was Trimethyl(methylcyclopentadienyl)-platinum(IV), (C_9_H_18_Pt). The EBID process was performed using a working distance of 5 mm, an electron voltage of 17.8 kV, and a beam current of 0.40 nA. The background vacuum of the system was 8.82 × 10^−7^ mbar and the precursor gas flux was adjusted such that the total pressure was 2.33 × 10^−6^ mbar, after which the EBID process was started. Single-dot exposure was used as the writing strategy, using stream files generated through MATLAB (MathWorks, Natick, MA, USA) scripts.

The resulting nanopatterns were characterized by scanning electron microscopy (SEM) performing using the same equipment. The height and base diameter were measured for thirty different pillars per sample using 52° tilted SEM images. The center-to-center spacing was also measured from the top view images. The dimensions of the produced nanopatterns are reported as mean ± standard deviation (Table 1).

Given that the static water contact angles could not be measured directly (due to the small size of the patterned areas), we used the Cassie-Baxter wettability model to estimate the values corresponding to different designs [28,29]. To this aim, the contact angle of an EBID-fabricated Pt-C layer was measured by a DSA 100 drop shape analyzer (Krüss, Hamburg, Germany) using deionized water. A volume of 2 µL liquid with a falling rate of 1667 µL min^−1^ was placed on the surface and the average contact angle was recorded within 30 s after the droplet touched the surface. The measured contact angle was further used to calculate the Cassie-Baxter contact angle (Table 1) [25].

Furthermore, topography images of the nanopatterns were acquired in Quantitative Imaging (QI) mode using an AFM JPK Nanowizard 4 (Berlin, Germany) and a high aspect ratio probe (TESPA-HAR, Bruker, Germany). A set point of 20 nN, a Z length of 300 nm, and a pixel time of 10 ms were used as scanning parameters. The images were analysed by using the JPK SPM data processing software (JPK instruments, v6.1, Berlin, Germany) to obtain 3D images of the surface and the average roughness.

### 2.2. Preparation of Bacterial Cultures

Gram-positive bacteria *Staphylococcus aureus* (RN0450 strain) (BEI Resources, VA, USA) was grown on brain heart infusion (BHI) (Sigma-Aldrich, MO, USA) agar plates at 37 °C overnight. A pre-culture of bacteria was prepared by inoculating a single colony in 10 mL autoclaved BHI, shaken at 140 rpm at 37 °C. The bacterial cells were collected at their logarithmic stage of growth and their optical density at 600 nm wavelength (OD_600_) in the medium solution was adjusted to a value of 0.1 to be finally cultured on the specimens. Such OD is equivalent to 148 × 10^6^ colony forming units (CFUs) per milliliter.

### 2.3. Investigation of Bactericidal Properties

Two independent sets of experiments were performed to evaluate the bactericidal properties of the nanopatterns. In each set of experiments, the nanopatterned areas of each specimen and the surrounding flat areas (Figure 1d) were considered as the study and control groups (all in triplicates) in the bacterial studies, respectively. The specimens were initially sterilized by through immersion in 70% ethanol and were exposed to UV light for 20 min prior to inoculation with 1 mL of the bacterial suspension in a 24-well plate (Cell Star, Germany). The specimens were then incubated at 37 °C for 18 h.

As previously explained [25], fabricating large areas of nanopatterned surfaces using EBID is not time-efficient yet, but patterning a small area would suffice for systematically studying the bactericidal effects of highly controlled nanopatterns. However, it hinders the use of certain assessment methods such as live/dead staining and CFU counting. We therefore exploited a method applied previously in several other studies [25,30,31,32,33] in which morphological evaluation of bacterial cells through SEM imaging is used to distinguish between disrupted and healthy bacterial cells. The validity of this technique has been demonstrated before in studies that have used and compared different evaluation techniques in determining the bactericidal properties of nanopatterned surfaces [30,33]. Therefore, to determine the killing efficiency of the nanopatterns, we first washed our specimens with phosphate-buffered saline (PBS) to remove any non-adherent bacteria. The adhered bacterial cells were then fixed by immersion in a PBS solution containing 4% formaldehyde (Sigma-Aldrich, St. Louis, MI, USA) and 1% glutaraldehyde (Sigma-Aldrich, St. Louis, MO, USA) at 4 °C for 1 h. Subsequently, the samples were dehydrated by a series of ethanol washing (50%, 70%, and 96% ethanol, respectively) and finally with hexamethyldisilazane (HMDS) (Sigma-Aldrich, St. Louis, MO, USA) for 30 min. After being air-dried, a thin layer of gold was sputtered on the specimens prior to being imaged by SEM at different magnifications and tilt angles of 0° and 52°. The killing efficiency was defined as the ratio of damaged cells to the total number of cells on the intended areas. Moreover, counting the total number of the bacterial cells attached to the nanopatterned and flat areas of each specimen enabled a comparison between the cell adhesion within the flat and nanopatterned surfaces.

### 2.4. Investigation of Nanopattern-Bacteria Interface

In order to further investigate the interactions between the bacterial cells and the nanopillars, and to analyse the possible killing mechanism of the nanopattern with the highest killing efficiency, focused ion beam scanning electron microscopy (FIB-SEM, FEI, Helios Nano Lab 650, OR, USA) was performed to acquire a cross-sectional view of the interface between cells and patterns. The specimen was tilted to 52°, at which angle the surface was milled using Gallium ions with a 7.7 pA ion beam (Z = 1.5 μm, operating voltage = 30 kV).

### 2.5. Statistical Analysis

To determine the statistical significance of the differences between the means of different experimental groups in terms of their bacterial cells attachment, a two-way ANOVA test was performed, followed by a Sidak’s multiple comparisons test, which was performed using Prism version 8.0.1 (GraphPad, San Diego, CA, USA). Similarly, the killing efficiency of different nanopatterns was statistically analyzed using the Mann-Whitney test. A *p*-value below 0.05 was considered to indicate statistical significance.

## 3. Results

### 3.1. Characteristics of the Fabricated Nanopatterns

Nanopillar arrays with different types of arrangements and interspacing values were successfully fabricated in 20 × 20 µm^2^ areas on each sample (Figure 1). Notwithstanding some slight variations, the intended dimensions were achieved for all the experimental groups (Table 1). The differences in the dimensions and arrangements of the nanopillars were clearly observed in SEM images (Figure 1) and AFM images (Figure 2). The average roughness decreased by increasing the interspacing of both ordered nanopillars (from 46.8 ± 4.9 nm for 100 *O* to 39.1 ± 1.8 nm for 500 *O*) and disordered nanopillars (from 68.3 ± 4.9 nm for 170 *D* to 24.8 ± 2.3 nm for 500 *D*). The density of the nanopillars decreased by increasing the interspacing, from 100.6 pillars per µm^2^ for 100 *O* to 4.5 pillars per µm^2^ for 500 *O* and 500 *D*. The nanopatterned surfaces were hydrophobic, as estimated by using the Cassie-Baxter model, with water contact angles ranging from 154° to 176°.

### 3.2. Effect of Interspacing on Bactericidal Properties

Before analyzing the effects of the selected parameters on the bactericidal activity of the nanopatterns, the bacterial adhesion to the specimens was studied to see whether any of the fabricated nanopatterns impairs the attachment of bacterial cells. Although care was taken to ensure seeding was as homogeneous as possible, there might be local differences in the number of the bacteria attached to the specimens. However, the ratio of the number of cells attached to the nanopatterned surfaces to those attached to a similar-sized flat area did not significantly vary between the specimens, meaning that bacterial cell adhesion was similar between all experimental groups (Figure 3a).

*S. aureus* cells showed their typical coccoid-shaped morphology on the flat areas with no significant sign of irregularity, disruption, or death (Figure 3b). On the contrary, ruptured bacterial cells with squashed morphologies were identified on the nanopatterned areas and were marked as damaged/dead cells (Figure 3f–i). The ordered nanopatterns with larger interspacing values (i.e., 300 *O* and 500 *O*) displayed significantly lower bactericidal efficiencies against *S. aureus* (8.6 ± 4.2% and 3.7 ± 2.3%, respectively) as compared to 100 *O* and 170 *O* (*p* < 0.01) (Figure 4a). While there was a significant difference in the killing efficiency between 300 *O* and 500 *O* (*p* < 0.05), no significant differences were observed between 100 *O* and 170 *O*.

### 3.3. The Effects of Controlled Disorder on Bactericidal Properties

The effects of controlled disorder on the bactericidal activity of the nanopatterned surfaces were evaluated by comparing the killing efficiency of the ordered and disordered nanopatterns with the same interspacing (Figure 3c–e). Although somewhat higher values of killing efficiencies were observed for the disordered nanopatterns with larger interspacing values (i.e., 300 *D* and 500 *D*) as compared to their ordered counterparts, the differences were not statistically significant (Figure 3b). The disordered nanopattern with the lower interspacing (i.e., 170 *D*) showed the same bactericidal efficiency as the ordered counterpart (Figure 4b).

### 3.4. Nanopattern-Bacteria Interface

Cross-sectional views showed that nanopatterns could penetrate the bacterial cell wall and cause their death by disrupting it (Figure 5). Moreover, the bending of the nanopillars underneath the bacterial cells (Figure 3 and Figure 5) suggested significant amounts of reciprocal forces that cells and pillars exert on each other.

## 4. Discussion

The main contribution of this study was shedding light on how interspacing and controlled disorder could affect the bactericidal properties of nanopatterns. The height and diameter of the nanopatterns produced were, therefore, kept constant while the interspacing and the arrangement of nanopillars were systematically altered in seven different study groups.

In order to elucidate the effects of these two parameters on the bactericidal properties of nanopatterns, it is crucial to first consider their killing mechanisms. The main killing mechanism of nanopatterns is widely believed to be the direct penetration of high aspect-ratio nanopatterns into the bacterial cell wall and disrupting it by exerting a high enough force [20,31]. Since the thickness and composition of cell walls are different in Gram-negative and Gram-positive bacteria [34], the same force will not rupture different cell walls equally [16,35]. Moreover, other factors such as hydrostatic and gravitational forces should be also considered when studying the interactions between bacteria and nanopatterns [36].

The results of this study showed that the cell wall of *S. aureus* could be mechanically penetrated by nanopatterns (Figure 5) and be severely damaged with an unrecognizable morphology (Figure 3), as reported previously [25,30,37]. However, changing the interspacing of nanopatterns could drastically affect the percentage of the damaged cells. Although aspect-ratio is a crucial design parameter of bactericidal nanopatterns, this effect seems to overshadow the effects of the nanopatterns aspect-ratio, which is corroborated by other studies in the literature. Linklater et al. showed that nanopillars with a diameter of 80.3 nm, an interspacing of 99.5 nm (similar to 100 *O* in the present study), and a height of around 430 nm (much higher than 100 *O*), exhibit a comparable level of bactericidal activity against *S. aureus* [38]. Similarly, another study [39] showed that naturally occurring nanopillars with the approximate height of 430 nm, the approximate tip diameter of 48 nm, and an interspacing of around 116 nm (i.e., larger height than 100 *O* but comparable interspacing), exhibit a killing efficiency of 39.4 ± 20.3% against *S. aureus* after 18 h. Computational simulations have also demonstrated that, as compared to the height, the interspacing has a substantially greater effect on the bactericidal properties of nanopillars [40]. It is plausible that a smaller interspacing and a higher density of nanopatterns result in more contact points between the bacteria and nanopatterns. This, in turn, leads to more physicomechanical interactions and higher chances of bacteria being ruptured [41], however, further studies are required to determine the minimum number of contact points that exert enough force to rupture the cell wall of different types of bacteria. According to the literature, the majority of bactericidal nanopatterns have an interspacing below 300 nm [20]. The reported values make more sense when it comes to bactericidal activity against *S. aureus*, which has a coccoid shape with a diameter larger than 500 nm [42]. For an interspacing exceeding the diameter of *S. aureus*, it is likely that bacterial cells land in between the nanopillars, thereby escaping the deadly spikes (Figure 3h,i).

An increased number of contact points for the lower values of interspacing could also contribute to some other proposed killing mechanisms. Bandara et al. [43] have argued that nanopillars do not directly interact with the bacterial cell membrane and showed that bacterial cells attach to surface nanopatterns via the expression of extracellular polymeric substances (EPS). Further movement of the bacteria on the surface on the one hand and strong EPS-mediated adhesion forces on the other lead to the stretching of the cell membrane beyond its rupture point. Considering this theory, an increased number of adhesion sites between surface features and expressed EPS could potentially amplify such a stretching mechanism. Although the bending of the nanopillars underneath the bacterial cells could be due to that movement rather than the bacteria weight only, further evaluations such as live imaging for real-time tracking of the bacteria on the surface, are required to confirm such a mechanism. Additionally, AFM measurements in another study have shown that the adhesion force of an EPS-producing *S. aureus* strain attached to nanopatterns does not significantly change due to differences in interspacing [44]. Altogether, the results of the present study are more consistent with the direct penetration theory, as it can be seen that nanopatterns have actually intruded about 101 nm (SD 9 nm) (Figure 5) into the bacterial cells, which is much larger than the cell wall thickness of *S. aureus* (i.e., 10–20 nm) [45].

Deviation from an ordered arrangement in nanopatterns has been shown to effectively influence the differentiation of the human mesenchymal stem cells [22] and the response of osteoblast cells [21]. Biochemical mechanotransduction pathways have been shown to be involved in translating mechanical cues (e.g., disordered nanopatterns) into biochemical responses (proteomic changes) [46]. An intricate network of cellular components is associated with receiving, transducing, and interpreting the mechanical cues [14]. In contrast, the bacterial mechanotransduction mechanisms seem to be simpler since the structure and components of bacterial cells are not as complex as those of mammalian cells [15]. Moreover, the information over the influence of nanopatterns disorder on the bacterial mechanotransduction is scarcely available in the literature. Further studies are, therefore, needed to elucidate the possible mechanobiological pathways pertaining to the interactions of bacteria and nanopatterns. Similar to another study that used *E. coli* [24], our results showed no significant difference between the bactericidal efficiencies of ordered and disordered nanopatterns against *S. aureus* as long as the interspacing is kept constant. Given the killing mechanism observed in this study, it can be concluded that introducing controlled disorder either does not change the mechanical force that bacteria experience or does not increase it beyond the threshold required for cell wall rupture. The findings of this study indicate that interspacing is a highly promising design parameter that should be further optimized to achieve nanopatterned surfaces with the highest potential of bactericidal activity. In combination with controlled disorder, nanopatterns with contrary effects on bacterial and mammalian cells may be developed in order to achieve selective biocidal activity.

## 5. Conclusions

The effects of interspacing and controlled disorder, as two design parameters, on the bactericidal properties of nanopatterns were systematically studied. Nanopatterns with constant heights and diameters of 190 nm and 80 nm, respectively, and different values of interspacing (i.e., 100, 170, 300, and 500 nm) were fabricated using EBID. A controlled disordered version of the nanopatterns with interspacings of 170, 300, and 500 nm was also fabricated. Quantifying the number of damaged *S. aureus* cells cultured on the nanopatterns for 18 h, showed that decreasing the interspacing significantly increased the bactericidal efficiency and the nanopatterns with 100 nm interspacing exhibited the highest efficiency (62.3 ± 23.1%). The nanopatterns with controlled disorder did not enhance the bactericidal efficiency compared with the ordered counterparts. Moreover, the direct penetration of nanopatterns into the bacterial cell wall and its eventual rupture due to the forces applied to it was shown to be the dominant killing mechanism of these nanopatterns which is consistent with the majority of the previous studies. Further research is required to elucidate whether interspacing and controlled disorder could affect the biochemical mechanotransduction pathways in bacteria.

## Figures and Tables

**Figure 1 nanomaterials-10-00347-f001:**
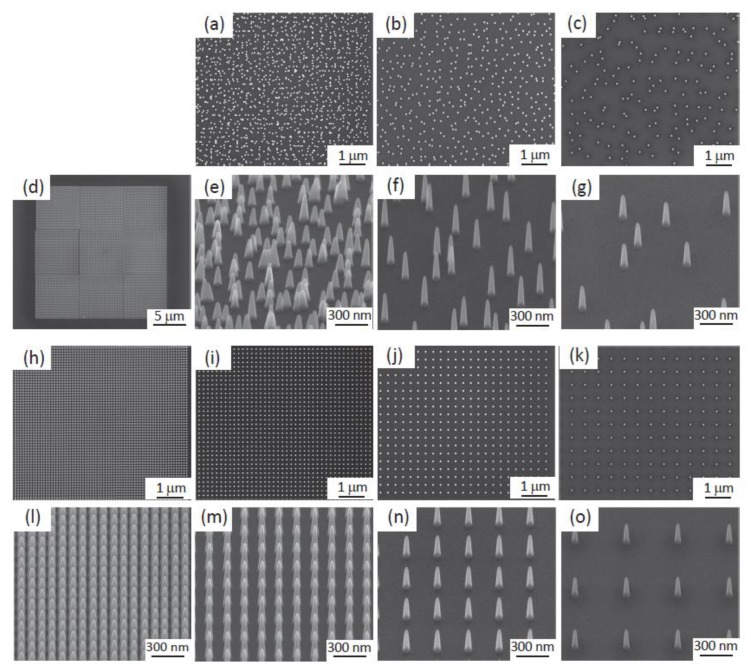
The scanning electron microscope (SEM) images of (**a**) the top view of the nanopillars with a height of 190 nm, a diameter of 80 nm, and an interspacing of 170 nm with 45 nm disorder distance (170 *D)*; (**b**) the top view of the nanopillars with an interspacing of 300 nm and 110 nm disorder distance (300 *D*); (**c**) the top view of the nanopillars with an interspacing of 500 nm and 210 nm disorder distance (500 *D*); (**d**) 20 × 20 µm^2^ nanopatterned areas on a Si substrate; (**e**) the tilted view of 170 *D*; (**f**) the tilted view of 300 *D*; (**g**) the tilted view of 500 *D*; (**h**) the top view of the ordered nanopillars with an interspacing of 100 nm (100 *O*); (**i**) the top view of the ordered nanopillars with an interspacing of 170 nm (170 *O*); (**j**) the top view of the ordered nanopillars with an interspacing of 300 nm (300 *O*); (**k**) the top view of the ordered nanopillars with an interspacing of 500 nm (500 *O*); (**l**) the tilted view of 100 *O*; (**m**) the tilted view of 170 *O*; (**n**) the tilted view of 300 *O*; (**o**) the tilted view of 500 *O*.

**Figure 2 nanomaterials-10-00347-f002:**
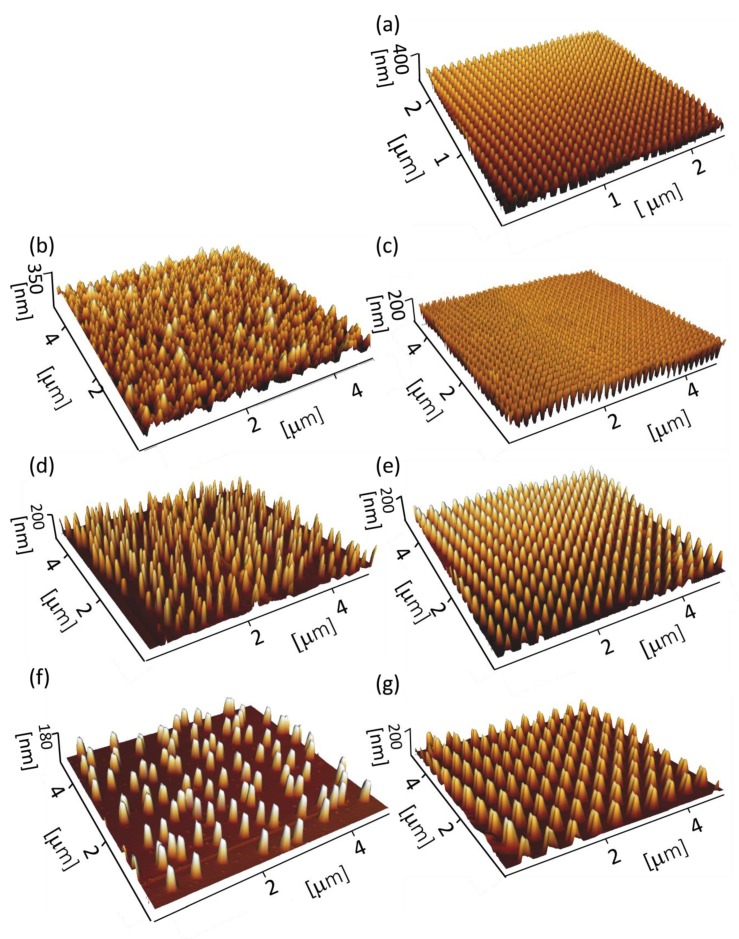
3D AFM images of (**a**) 100 *O*; (**b**) 170 *D*; (**c**) 170 *O*; (**d**) 300 *D*; (**e**) 300 *O*; (**f**) 500 *D*; (**g**) 500 *O* nanopillars.

**Figure 3 nanomaterials-10-00347-f003:**
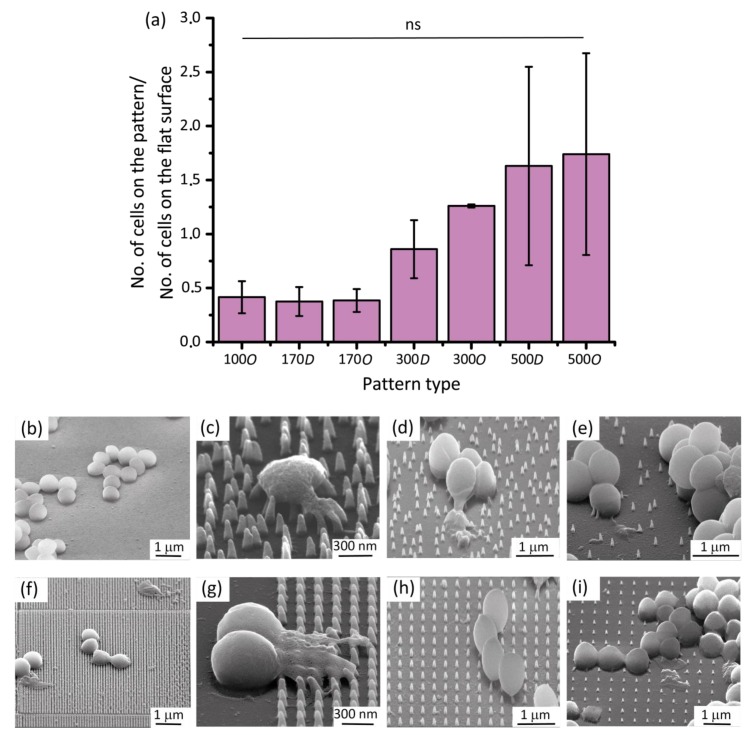
(**a**) The ratio of the bacterial cells attached to the nanopatterns to those attached to similar flat areas. No significant difference in that ratio for different types of nanopattern indicates that bacterial adhesion is similar between different groups; The SEM images of *S. aureus* bacteria on (**b**) the control Si surface; (**c**) 170 *D*; (**d**) 300 *D*; (**e**) 500 *D*; (**f**) 100 *O*; (**g**) 170 *O*; (**h**) 300 *O*; (**i**) 500 *O* at 52° tilted view. The damaged bacterial cells can be identified with irregular and unrecognizable morphologies as compared to normal cells on flat surfaces.

**Figure 4 nanomaterials-10-00347-f004:**
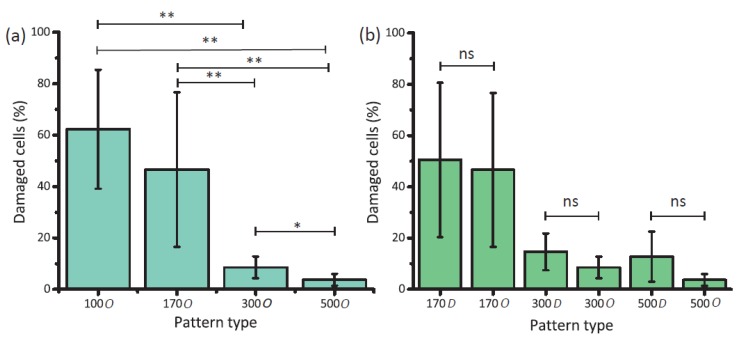
(**a**) The effects of interspacing on the bactericidal efficiency of nanopatterns (* *p* < 0.05 and ** *p* < 0.01); (**b**) The effects of controlled disorder on the bactericidal efficiency of nanopatterns. No significant differences were observed between the ordered and disordered nanopatterns but the value of interspacing.

**Figure 5 nanomaterials-10-00347-f005:**
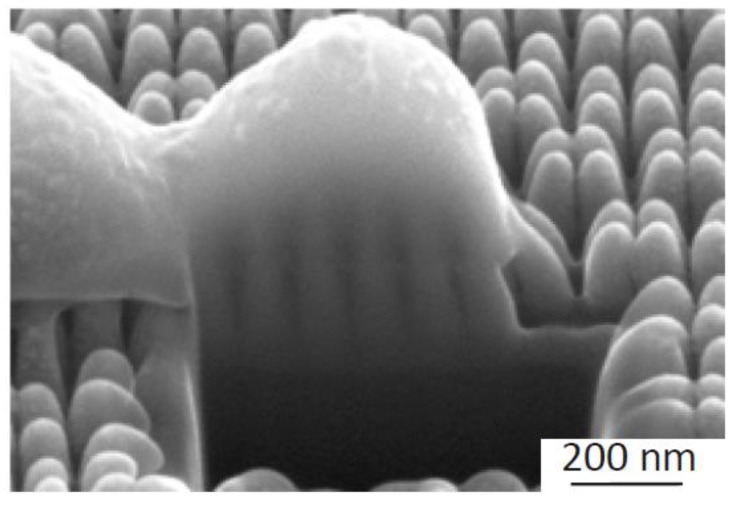
The SEM image of the interface between bacteria and the 100 *O* nanopattern. The depth of direct penetration of the nanopillars into the bacterial cell wall (about 100 nm) far exceeds the average cell wall thickness of *S. aureus* (i.e., 10–20 nm).

**Table 1 nanomaterials-10-00347-t001:** The characteristics of the nanopatterns produced by EBID.

Pattern Type	Interspace (nm)	Disorder Distance (nm)	Height (nm)	Base Diameter (nm)	Aspect Ratio	Pillar Density (Number/µm^2^)
100 *O*	100	0	198 ± 7	89 ± 4	2.2 ± 0.0	100.6
170 *D*	170	45	200 ± 13	73 ± 9	2.7 ± 0.2	35.5
170 *O*	170	0	188 ± 22	67 ± 13	2.8 ± 0.2	35.5
300 *D*	300	110	177 ± 13	73 ± 3	2.4 ± 0.1	11.9
300 *O*	300	0	190 ± 28	71 ± 2	2.7 ± 0.3	11.9
500 *D*	500	210	201 ± 15	76 ± 4	2.6 ± 0.1	4.5
500 *O*	500	0	184 ± 16	72 ± 3	2.6 ± 0.1	4.5

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
