# Peer review of "Deciphering the Roles of Interspace and Controlled Disorder in the Bactericidal Properties of Nanopatterns against Staphylococcus aureus"

_nanomaterials, 2020, doi:10.3390/nano10020347_

Round 1

Reviewer 1 Report

Recently, bactericidal effect on the nanotexture has been attracted attention because the effect is occurred by physical interaction between the bacteria and nano-patterned surface. The authors reported that bactericidal effect depends strongly on the interspace between nano-pillars. Nano-pillars composed of Pt/C which diameter about 80 nm and height about 200 nm were fabricated on flat Si substrate by using electron beam induced deposition (EBID). Then, Interspace between them were changed from 100 to 500 nm with or without disorder of pillars. The obtained results demonstrated that bactericidal effect depended strongly with the interspace between nano-pillars when S. aureus was used as an example of gram-positive bacterium. It is well known that bactericidal effect on the nanopatterned surface is due to the interaction between the bacteria and its surface. Thus, it is noted that bactericidal effect is different between the species. In addition, other research group (M. Michalska et al., Nanoscale, 2018, pp.6639-6650) has reported that bactericidal effect depends on the interspace between nano-pillars and height of them. The authors should describe the advantage of this report comparing with the above paper. In my opinion, the paper is accepted after the major revision. My comments and questions are listed below.

The authors should change the title to concrete one. The current title seems rough. For example, “Bactericidal properties for aureus dependent with interspace of nano-pillars.” Why did the authors use only gram-positive bacteria? The authors had reported (ref. 25) that nano-pillar arrays fabricated by EBID showed the bactericidal property to gram-positive and gram-negative bacteria, which was also written in L.74-78. How did the authors decide the disorder distance? In addition, why there were no condition of disorder distance on the interspace of 100 nm. The authors should clarify the reasons. How did the authors decide the size of nano-pillars which aspect ratios were around 2.5. The authors should clarify the reasons. In Fig.2(a), I recommend that the vertical axis shows the number of adhered cells per unit area. I don’t understand why the authors standardized by the number of adhered cells on the flat surface. The most important question is how the authors count the number of damaged cells. How did author determine attachments on the surface derived from deformed cells, dust or carcass which were contaminated primary. I think that we can’t identify whether the attachments on the nano-pillar surface are derived from deformed cells by using SEM only. The authors should clarify the way to determine the attachments on the surface. Did the author count the deformed cells on each area of 20x20 um2? Did the author use FIB or other equipment to have cross sectional piece for Fig.4? I wonder why the front side of the piece was cut.

Reviewer 2 Report

1, This study was shedding light on how interspacing and controlled disorder could affect the bactericidal properties of nanopatterns, Which have contributed a lot on the future studies aimed at optimizing the design of nanomaterials. However, I think that the authors have insufficient experimental data on the surface characteristics, which cannot fully explain the difference between the ordered nanopatterns and disordered nanopatterns surfaces on the mechanism of killing the bacteria. Please add more experiments related to surface characteristics, such as surface contact angles, roughness, SPM etc.

2, In the section on 2.3. Investigation of bactericidal properties, please provide the size of the area selected when calculating the dead bacteria rate. Please provide a larger area SEM image in the Figure 2.

Reviewer 3 Report

This is a really clear paper that tests a simple hypothesis concerning the areal density and order/disorder of nanopillars on their bactericidal properties.  The paper is well written, results are significant and I think will generate significant interest. It is a good fit with the special issue on antimicrobial nanomaterials.

Very minor points to address: For the non-biological scientists, it would be helpful to define abbreviations such as PBS and CFU.  The source of the error bars in figure 3 was not obvious to me.  (Is it 2 standard deviations?)

I struggled a little with the interpretation of figure 2.  It looks to me that the proportion of cells adhered to a surface relative to the flat surface increases with increasing interspacing, but that the proportion of cells adhered is independent of the patterning (ordered or disordered).

If the discussion could spell out the literature conclusions in the prior art in a little more detail, it would help to increase the impact of this work.  If aspect ratio of the surface features does not matter, does the length of the feature matter.  Furthermore, a simple calculation of the approximate number of 'spikes' that the bacterial cell must sit on for each areal density would be interesting.  I'd love to know if there is something significant about whether each bacterium is ok to sit on one spike, but not 2, or if there is a cut-off at some other point.

Although the effect of order/disorder is likely to be the most original part of this study and it reveals little effect, the fact that this might differentiate between different cell types could be a useful design criterion in achieving selective toxicity. 

Round 2

Reviewer 1 Report

The authors have revised the manuscript according to the refrees' comments. Then, the paper is acceptable for the publication without any corrections.